# The impact of Connexin 43 deficiency on the cell shape and cytoskeleton of murine Sertoli cells: A house with ramshackle walls?

Mareike Ueffing[1], Marion Langeheine[1], Sarah Gniesmer[1]*, Kristina Rode[1], Sarah Staggenborg[1], Gudrun Wirth[1], Kerstin Rohn[2], Rüdiger Koch[1], Ines Blume[1], Christiane Pfarrer[1], Christoph Wrede[3], Ralph Brehm[1]

1 Institute for Anatomy, University of Veterinary Medicine Hannover, Foundation, Hannover, Germany,
2 Department of Pathology, University of Veterinary Medicine Hannover, Foundation, Hannover, Germany,
3 Research Core Unit Electron Microscopy, Institute of Functional and Applied Anatomy, Hannover Medical School, Hannover, Germany

* sarah.gniesmer@tiho-hannover.de

## Abstract

Genetically induced loss of the gap-junction protein Connexin 43 (Cx43) in murine Sertoli cells leads to an arrest of spermatogenesis at the level of spermatogonia, highly vacuolated tubules, and intratubular cell clusters. Transmission electron microscopy as well as 3D-reconstruction of Sertoli cells based on serial block-face scanning electron microscopy imaging revealed severe cell shape changes in Cx43 deficient Sertoli cells. Since the cytoskeleton is important for the transport of germ cells within the seminiferous epithelium and for keeping the cell shape, the study at hand aimed to reveal correlations of Cx43 loss and changes of cytoskeletal components and their spatial organization in the seminiferous epithelium. Immunohistochemistry, immunofluorescence, conventional transmission electron microcopy and immunogold labeling indicated alterations in microtubule and actin filament distribution patterns in Cx43 deficient Sertoli cells compared to wildtype mice. Firstly, microtubules seemed to be misoriented in mutant Sertoli cells. Secondly, the actin filament based basal ectoplasmic specializations were increased in spatial extension, but the apical ectoplasmic specialization was missing. Lastly, Sertoli cells of both genotypes immunostained positive for vimentin, the prevalent intermediate filament of Sertoli cells, but not for keratins, markers for Sertoli cell immaturity or dedifferentiation. In conclusion, Cx43 deficiency in Sertoli cells correlates not only with severe cell shape alterations but also with changes in microtubule and actin filament distribution patterns, while intermediate filament expression seems to be only negligibly influenced.

**Data availability statement:** All relevant data are within the paper and its Supporting Information files.

**Funding:** The author(s) received no specific funding for this work.

**Competing interests:** The authors have declared that no competing interests exist.

## 1. Introduction

An arrest of spermatogenesis at the level of spermatogonia and certain human neoplastic testicular diseases (e.g., seminoma and the germ cell neoplasia in situ, formerly known as carcinoma in situ) correlate with the absence or down-regulation of the gap-junction protein Connexin 43 (Cx43) [1–5]. Hence, a conditional knockout (KO) mouse model was supposed to reveal the specific roles of Cx43 in Sertoli cell (SC) function. Since a global KO of the encoding gene (*Gja1*) turned out to be fatal due to cardiac malformation in neonatal mice [6], an SC specific knockout mouse model was generated. Homozygous KO mice (SCCx43KO$^{-/-}$) were infertile despite unvaried libido. Total and relative testis weights were significantly lower compared to their wild type (WT) littermates. Microscopic investigation of testis samples disclosed a dramatic germ cell (GC) deficiency and an arrest of spermatogenesis at the level of spermatogonia or even a Sertoli cell-only syndrome (SCOS). Only single tubules showed residual spermatogenesis. Additionally, these Cx43 deficient mice featured intratubular SC clusters and severe SC vacuolization [7,8] (S1 Fig). Subsequently, many efforts were already made to further understand how these histological alterations can be traced back to Cx43 channel or Cx43 non-channel functions in SC and its loss.

In the face of SC that seem to be unable to support spermatogenesis, the blood-testis-barrier (BTB) springs to mind, especially since Cx43 was found to form an integral component of SC-SC-junctional complexes and was supposed to be a regulator of the BTB function (opening and closing) [9]. The BTB is a unique barrier that "ties" the SC to adjacent SC. Thereby, it creates an immunological barrier that segregates the events of meiosis and postmeiotic GC development from systemic blood circulation. The production of "anti-sperm antibodies" is thereby prohibited [10]. Surprisingly, the BTB of SCCx43KO$^{-/-}$-mice appeared to be morphologically intact and unvaried in density. By virtue of an altered expression of adherens- and tight junction proteins like N-cadherin, β-catenin, occludin, claudin-11 and ZO-1, a disturbed dynamic of the opening and closing mechanism of the BTB can be presumed [11–13].

Impairment of SC functions (and spermatogenesis) could also derive from altered development, differentiation or maturation. A prolonged period of anti-Müllerian hormone (AMH) synthesis [14] and a partial disruption of the androgen receptor signaling pathway [15] underpin this theory, but still offer no satisfying explanation for the phenotypically altered cell shape of Cx43 deficient SC.

The present manuscript aims to shed light on another possible explanatory approach for the histological/morphological alterations of Cx43 deficient SC: the impact of Cx43 deficiency on the SC cytoskeleton.

### 1.1. Why the cytoskeleton?

The cytoskeleton plays a crucial role in determining the cell shape of somatic SC, SC polarity, intracellular transport and the positioning of cell organelles, establishment of the BTB as well as transport and anchoring of developing GC [16]. The

cytoskeletal system consists of three types of protein fibers: microtubules, actin filaments and intermediate filaments. These cytoskeletal structures organize into networks that serve the cell as a scaffold comparable to the posts and beams of a building. However, unlike a building's architecture, the cytoskeletal system is constantly being remodeled in response to externally applied forces or cellular processes. Assembly and disassembly are controlled by a plethora of associated proteins [17]. All of that is a requisite for dynamic processes within the SC (chiefly the support of spermatogenesis).

**1.1.1. Microtubules.** Adult SC are columnar cells extending from the basement membrane of the seminiferous epithelium to the lumen. The stem-like cells develop three-dimensional processes around differentiating GC like the twigs and branches of hardwood trees [18]. Especially microtubules are important for the maintenance of this unique cell shape [19,20]. Microtubules are hollow tubes formed by polymerization of subunits called α- and β-tubulin [21]. Normally, these polar tube-shaped filaments are linearly positioned in parallel to the long axis of the SC [16,18]. That arrangement enables microtubules (in collaboration with the motor proteins dynein and kinesin) to serve as tracks for the transport of elongating spermatids along the apico-basal cell axis. Finally, microtubules facilitate spermiation at the luminal edge of the seminiferous tubule [16,22].

**1.1.2. Actin filaments.** Filamentous actin (F-actin) is a two-chained helical polymer consisting of "globular" actin-monomers (G-actin) [21]. Actin filaments can be detected in nearly every region of the SC cytoplasm but are particularly dense and abundant in the cell periphery at sites of testis specific intercellular junctions [23]. These specific junctions enclose the ectoplasmic specializations (ES) [24] and the tubulobulbar complexes [25].

ES consist of (1) the SC plasma membrane, (2) a layer of hexagonally packed actin filaments, and (3) a cistern of endoplasmic reticulum [24,26]. ES occur at two different locations:

1. First, the **basal ES** are part of the BTB. Located close to the basement membrane, they are accompanied by tight junctions, desmosomes, and gap junctions [27].

2. Second, the **apical ES** serve as an anchoring device for developing spermatids within SC membrane niches. They surround the heads of elongating and elongated spermatids. Earlier staged GC are connected to SC via desmosome-like junctions [28].

Both basal and apical ES must be modified to allow the normal transit of preleptotene spermatocytes across the BTB or the release of elongated spermatids into the tubular lumen (spermiation). The second actin-based structures, the **tubulobulbar complexes**, are involved in this turnover [29]. These structures can be imagined as double membrane invaginations that internalize (and thereby abrogate) intercellular junctions [25].

In short, SC actin filaments are crucial for BTB function and remodeling, positioning and anchoring of GC within the seminiferous epithelium as well as for spermiation.

**1.1.3. Intermediate filaments.** The third component of an SC cytoskeleton are intermediate filaments. Intermediate filaments mainly surround the SC nucleus and send out extensions in apical cell regions [14,30–33]. In contrast to many epithelial cells, whose prevalent intermediate filaments are of the keratin type, adult SC primarily contain vimentin [34]. The keratins 8, 18, and 19 can only be found in fetal murine SC [31,35,36]. In these early stages of SC development, both keratins and vimentin are temporarily co-expressed before keratins gradually vanish and vimentin prevails [31]. However, in seminiferous tubules of men with spermatogenic impairment (such as spermatogenic arrest at the level of spermatogonia or SCOS) as well as SC of GC neoplasia in situ infiltrated tubules and in senile SC, keratins 8 and 18 are reexpressed in SC [4,37,38]. Altogether, the keratins 8, 18, and 19 are deemed to be associated with SC dedifferentiation or immaturity. Since the SCCx43KO$^{-/-}$ also leads to similar histological phenotypes as well as to SC with immature or abnormal features compared to human SC, an altered intermediate filament expression could be assumed. This seems likely since next-generation sequencing using testicular tissue of juvenile murine SCCx43KO$^{-/-}$-mice detected an upregulation of the *Krt 18* gene in prepubertal KO mice [39].

## 2. Materials and methods

### 2.1. Experimental animals and tissue sampling

SCCx43KO$^{-/-}$-mice were generated using the Cre/loxP recombinase system. Breeding strategy, animal housing, and confirmation of Cx43 gene loss via PCR and β-galactosidase immunohistochemistry (IHC) were realized as described by Brehm et al. 2007 [8].

For IHC, mice were chosen from postnatal day 2 up to adulthood (days in detail: 2, 5, 8, 10, 12, 15, 18, 31, 68, 100, 104, 134, 151, 176, 191, and 225 with n=1–3 per genotype and age group). For immunofluorescence (IF), one adult mutant mouse (205 days old) and one adult WT mouse (160 days old) were sacrificed. Two adult KO and two adult WT mice were sacrificed for TEM and another two adult KO mice and two WT littermates for immunogold labeling.

All husbandry and experimental procedures were approved by the Lower Saxony State Office for Consumer Protection and Food Safety, Oldenburg, Germany ([33.19-42502-05-16A017] and [33.8-42502-05-21A575]) as well as performed in accordance with the relevant guidelines and regulations, including ARRIVE (Animal Research: Reporting of In Vivo Experiments) guidelines.

All animals were housed in groups at room temperature of between 22 °C and 24 °C and a relative humidity of 60–65% with a 12-hour day-night cycle. The mice had free access to tap water and standard pellet food (rat/mouse - maintenance, ssniff Spezialdiäten GmbH, Soest, Germany) at all times.

### 2.2. Immunohistochemistry (IHC)

Animals were killed by cervical dislocation. Both testes were removed immediately and fixed in Bouin' s solution for 48 h. Samples were then gradationally dehydrated in ethanol and embedded in paraffin wax according to standard protocols. Four μm sections were cut and mounted on glass slides. During deparaffinization and rehydration, endogenous peroxidase was blocked using hydrogen peroxidase. Antigen retrieval was conducted by boiling slides for 20 min in buffer (Table 1) at 96–99 °C. Afterwards, sections were blocked using 3% bovine serum albumin (BSA) for 20 min. Primary antibodies against α- and β-tubulin, β-actin, vimentin, and keratins (Table 1) were diluted in 1% BSA and incubated overnight in a humidified chamber. The following day, slides were incubated with the corresponding horseradish peroxidase conjugated secondary antibodies (Table 1). Immunoreaction was visualiszed by diaminobenzidin (DAB; En Vision, Dako GmbH, Hamburg, Germany). Finally,

**Table 1.  Antibodies used for immunohistochemistry, immunofluorescence and immunogoldlabelling.**

| Antibody | Company | Dilution | Antigen Retrieval | Secondary Antibody |
|---|---|---|---|---|
| α-tubulin (11H10) Rabbit mAb | Cell Signaling Technology | 1:4000 | sodium citrate buffer (pH 6.0) | anti-rabbit En Vision; DAKO, Hamburg, Germany |
| β-tubulin, clone TUB 2.1 (T4026) | Sigma-Aldrich | 1:200 | sodium citrate buffer (pH 6.0) | anti-mouse En Vision; DAKO, Hamburg, Germany |
| β-actin (13E5) Rabbit mAB | Cell Signaling Technology | 1:200 and 1:1000 | sodium citrate buffer (pH 6.0) | anti-rabbit En Vision; DAKO, Hamburg, Germany |
| Claudin 11 (CAP1843) | Cell Signaling Technology | 1:2000 | | Alexa fluor 546 Rabbit IgG; Thermo Fisher Scientific, dilution 1:1000 |
| DAPI-dihydrochlorid (D9542) | Sigma-Aldrich | 1:500 | | |
| Fluoresceinisothiocyanat-Phalloidin (5282) | Sigma-Aldrich | 1:70 | | |
| Vimentin (C-20) sc-7557 | Santa Cruz Biotechnology | 1:200 | sodium citrate buffer (pH 6.0) | anti-rabbit En Vision; DAKO, Hamburg, Germany |
| Pan-Cytokeratin CK102 (AE1/AE3) | Biologo | 1:100 | TEC buffer (pH 7.8) | anti-mouse En Vision; DAKO, Hamburg, Germany |
| Pankeratin (C11) mouse mAb | Cell Signaling Techonolgy | 1:250 | EDTA buffer (pH 9.0) | anti-mouse En Vision; DAKO, Hamburg, Germany |

sections were dehydrated and mounted with Eukitt® (O.Kindler GmbH, Freiburg, Germany). Except for the omission of the primary antibody, the same protocols were used for the respective negative controls (S2 Fig). As positive controls for antibodies directed against keratins, Bouin-fixed sections of murine liver, kidney, skin and uterus were used.

### 2.3. Immunofluorescence (IF)

Testes were fixed in 3% paraformaldehyde (PFA) for 24 hours and than stored at -80 °C. Specimens were cut with a cryotome, air dried for 5 minutes, fixed again in 3% PFA for 8 min and in aceton for 5 min at -20 °C. After washing in phosphate buffered saline (PBS), treatment with 0.5% Tween (Tween 20, Carl Roth GmbH & Co., Karlsruhe, Germany) in PBS and blocking with 3% BSA for 30 min, sections were incubated overnight with an antibody against claudin-11 (Table 1). Afterwards, samples were incubated with the secondary antibody (Table 1) for 45 min, phalloidin (Table 1) for 60 min, and DAPI (Table 1) for 5 min. Finally, sections were mounted with ProLong® Gold Antifade Reagent (Thermo Fisher Scientific GmbH, Darmstadt, Germany, P36930).

### 2.4. Transmission electron microscopy (TEM)

Testes were fixed in Karnovsky's fixative (Na-cacodylate buffered 2.0% paraformaldehyde, 2.5% glutaraldehyde fixative, pH 7.4) and later cut into smaller pieces. After washing in 0.15 M sodium-cacodylate buffer (pH 7.4) en-bloc osmication was carried out according to the OTO-protocol (osmiumtetroxide, thiocarbohydrazide, osmiumtetroxide) with uranyl acetate and lead aspartate. Gradational dehydration in acetone was followed by Durcupan™ embedding (Durcupan™ ACM resin from Sigma Aldrich Corp., St. Louis, MO, USA). For at least seven days, polymerization was conducted at 60 °C. Areas of interest were defined based on semi-thin sections stained with 0.1% toluidine blue dye (0.1 g toluidine blue diluted in 100 mL distilled water and 0.25 g sodium hydrogen carbonate). Finally, resin blocks were trimmed, ultra-thin sections (60 nm) were cut, collected on grids and examined with a Zeiss EM 906 transmission electron microscope (Carl Zeiss Microscopy GmbH, Oberkochen, Germany).

### 2.5. Serial block-face scanning electron microscopy (SBF-SEM) and 3D-reconstruction

For 3D reconstruction of SC, already existing testicular probes and their SBF-SEM images were used. The exact procedure is described in the appropriate literature [40]. Here, only a brief description follows: resin blocks (prepared exactly like those for TEM) were mounted on aluminium specimen pins (Gatan Inc., Pleasanton, CA, USA), trimmed, and sputter-coated with a gold layer. In a Gatan 3View2XP system (Gatan Inc.), equipped Zeiss Merlin VP Compact SEM (Carl Zeiss Microscopy GmbH, Jena, Germany) automatic imaging of the blockface scanning was performed. For image processing, segmentation, and analysis, the freeware Microscopic Image Browser (MIB) was used [41]. Images were resampled from 10,000 x 10,000 pixels and 15 nm pixelsize to 2000 x 2000 pixels and 75 nm pixelsize as well as converted from 16 bit to 8 bit to provide a smaller, manageable dataset with still ample resolution. About 500 pictures were stacked. Single adult SC (n=3 per genotype), including the nucleus and possible adjacent GC as well as one intratubular SC cluster (consisting of 32 cells) were segmented. 3D reconstruction was conducted using 3dmod (IMOD 4.11.) [42].

### 2.6. Immunogold labeling

Karnowsky fixed testes were LR White embedded (modified from Skipper et al. 2008 [43]). Specimens were dehydrated in 30%, 50%, 70%, 90% and 100% ethanol at 4 °C for at least 30 min at a time. Incubation in a 1:2, 1:1, and 2:1 mixture of LR White and 100% ethanol, for 30 min each followed. Overnight the samples were incubated in pure LR White at 4 °C. After encapsulating, the specimens were incubated at 50 °C overnight to polymerize the resin. After having inspected the toluidine blue stained semi-thin section, blocks were trimmed and thin cuts (60 nm) were mounted on nickel grids. The grids were incubated in drops of 0.05 M glycin in PBS for 20 min and in drops of 5% BSA and 1% cold water fish skin

gelatine (CWFS) in PBS for 30 min. Incubation with the primary antibodies against α-tubulin, β-actin, or vimentin was conducted overnight at 4 °C. The same antibodies as listed in Table 1 were used, but factor 10 was less diluted. After several washing steps in 0.5% BSA and 0.1% CWFS in PBS, the grids were incubated with a secondary antibody (Goat-anti-Rabbit IGg (H&L) 10 nm gold labeled (810.011) from the Biotrend cliniSciences group, dilution 1:30) for 60 min, washed in PBS, fixated in 2% glutaraldehyde and tenderly counterstained with diluted uranyl acetate. Examination was conducted using a Zeiss EM 906 transmission electron microscope (Carl Zeiss Microscopy GmbH). Except for the omission of the primary antibody, the same protocols were used for the respective negative controls (S2 Fig).

## 3. Results

### 3.1. Cell shape

SFB-SEM images (exemplary videos in S3 and S4 Files) and 3D-reconstruction of WT and Cx43 deficient SC corroborated what light microscopy and TEM already suggested: while WT SC showed the characteristic stem-like cell-shape with cytoplasmic extensions around many differentiating GC, Cx43 deficient SC looked as if they were collapsed like deflated, soggy balloons (Fig 1B–1D and 1F–1H). Using TEM (2D image), mutant SC seemed to have numerous finger-like cell extensions which intertwine with cell extensions of the adjacent SC (S3 and S4 Files). 3D reconstruction revealed that this effect was mainly created because the Cx43 deficient cells were no longer strictly oriented towards the lumen like a tree trunk, but were rather deformed. Mutant SC were polymorphic, sometimes resembled a wave (Fig 1C and 1G), and sometimes they resembled a long-drawn star (Fig 1B and 1F). Within the cytoplasm, many Cx43 deficient SC featured vacuoles. These SC looked bulgy in 3D reconstruction like a snake that gobbled a large prey animal (video sequences of three KO SC and one WT SC in S3–S6 Files).

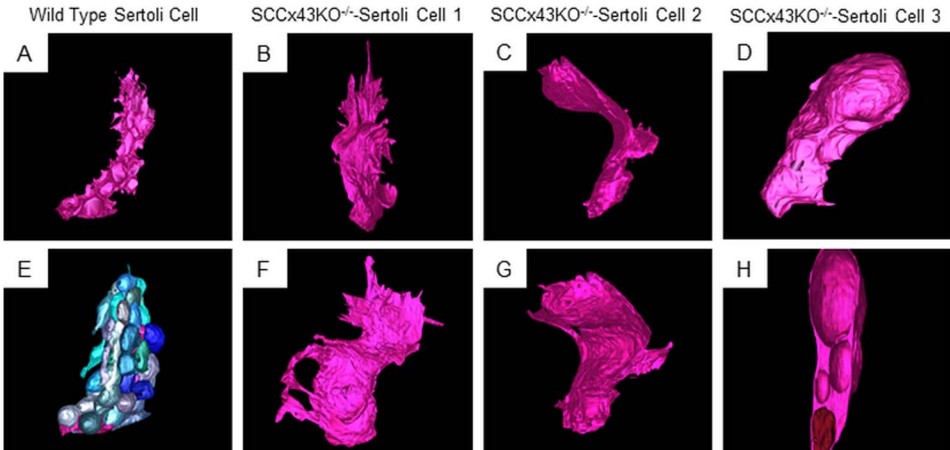

**Fig 1. 3D reconstruction of one wild type Sertoli cell without (A) and with germ cells (E) and three SCCx43KO⁻ᐟ⁻-Sertoli cells (B and F, C and G, D and H) from two different viewing angles.** The wild type Sertoli cell (pink) nurtures several germ cells (blue) from spermatogonia to spermatids (E). Different blue tones do not resemble different developmental germ cell populations, but rather are chosen to visually establish borders between the different germ cells. However, germ cells are situated in so-called "spermatogenic niches". Thoses niches are reminiscent of hemisphere-like sockets that "cave" in the Sertoli cell membrane (E). The Sertoli cell itself resembles a tree trunk that reaches from the basement membrane (bottom edge of the picture) to the lumen of the seminiferous tubule (upper edge of the picture). Connexin 43 deficient Sertoli cells are polymorphic and seem to be collapsed. They feature several strange finger-like cell extensions reminiscent of arms or spikes (B, F, and G). Some mutant Sertoli cells (here shown in SCCx43KO⁻ᐟ⁻-Sertoli cell 2 in C and G) exhibit wave-like shapes. Those cells are mostly located in the direct vincity of Sertoli cells that contain vacuoles (like SCCx43KO⁻ᐟ⁻-Sertoli cell 3 in D and H). Vacuoles containing Sertoli cells are bulged and beaten like a snake that swallowed a very large prey animal. Hence, vacuoles account for large parts of the cell. To visualize this, SCCx43KO⁻ᐟ⁻-Sertoli cell 3 (D) is "cut in half" (H). The basally located brown knob represents the Sertoli cell nucleus. The "holes" above the nucleus depict the vacuoles.

## 3.2. Microtubules

In adult WT mice, α-tubulin IHC revealed a radial, spoke-like staining pattern. The linear tubulin immunostaining was directed towards the tubular lumen and therefore ran along the longitudinal axis of the SC (Fig 2A). Dependent upon the stage of spermatogenesis, the staining pattern resembled either narrow cables, broad bands, or bands that branched out like trees in order to surround spermatids in the apical SC regions. A similar stage-specific staining pattern has been described in meticulous detail by Wenz et al. in rat SC immunostained with an antibody directed against tyrosinated α-tubulin [44].

In contrast, adult SCCx43KO⁻/⁻-SC showed a diffuse cytoplasmic immunoreaction for α-tubulin (Fig 2B) or many cable-like structures that were laterally distracted and not in line with the lumen. Somehow, the latter resembled the frayed ends of ropes that deviated to one side or the other. Additionally, the cable-like structures seemed to be more closely spaced than in WT mice. No typical spoke-like staining pattern was recognizable. Intratubular cell clusters, which are mainly formed by abnormal SC [14], were also diffusely immunopositive for α-tubulin.

Since microtubules are formed by polymerization of two subunits (α- and β-tubulin), β-tubulin IHC was performed on consecutive sections of the same mice. The used antibody was directed against all β-tubulin isoforms and caused the same staining pattern in WT and mutant mice as the antibody against α-tubulin (Fig 2C and 2D).

In addition, comparison of newborn SCCx43KO⁻/⁻-mice and their same aged WT littermates revealed no differences in α-tubulin distribution. In 2-, 5- and 8-day old mice, α- tubulin was more or less diffusely distributed in the cytoplasm of the

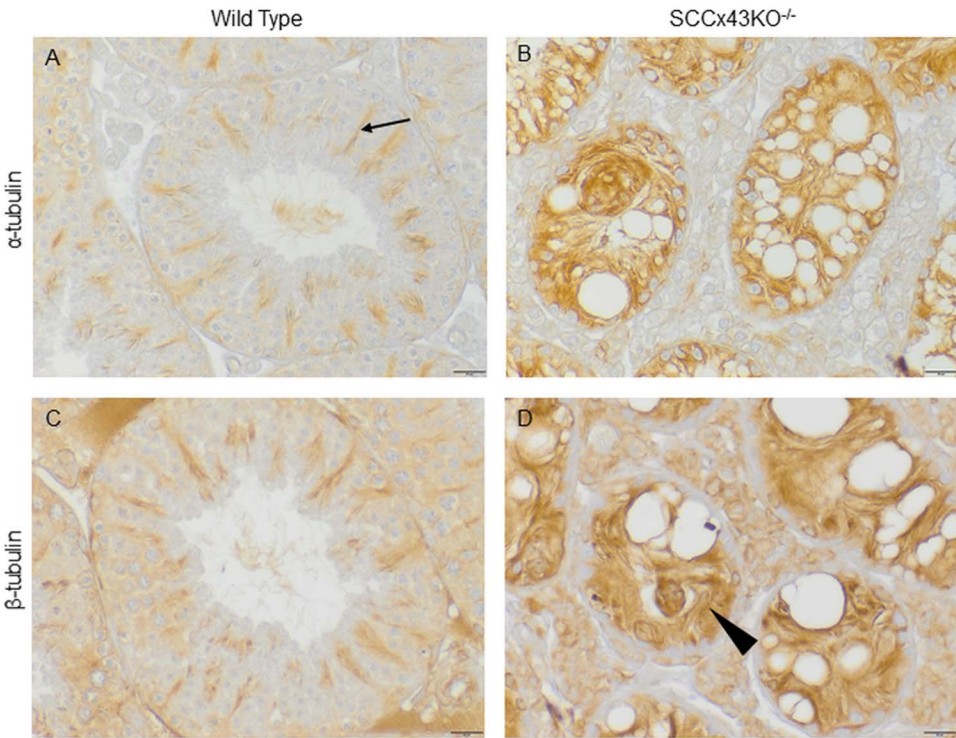

**Fig 2. Immunostaining for α- (A, B) and β-tubulin (C, D).** Seminiferous tubules of adult wild type mice (A, C) show a stage-specific spoke-like staining pattern similar to what is known for rats [44]. The radial staining pattern consists of band- or cable-like structures extending from the tubular base to the lumen (arrow). In some epithelial stages, the band-like structures branch out in apical Sertoli cell regions and surround spermatids (C). In contrast, Cx43 deficient Sertoli cells (B, D) show a rather diffuse cytoplasmic staining pattern of α-tubulin or many, tightly packed cable-like-structures that are laterally distracted and not in line with the lumen (arrowhead); scale bars = 20 μm.

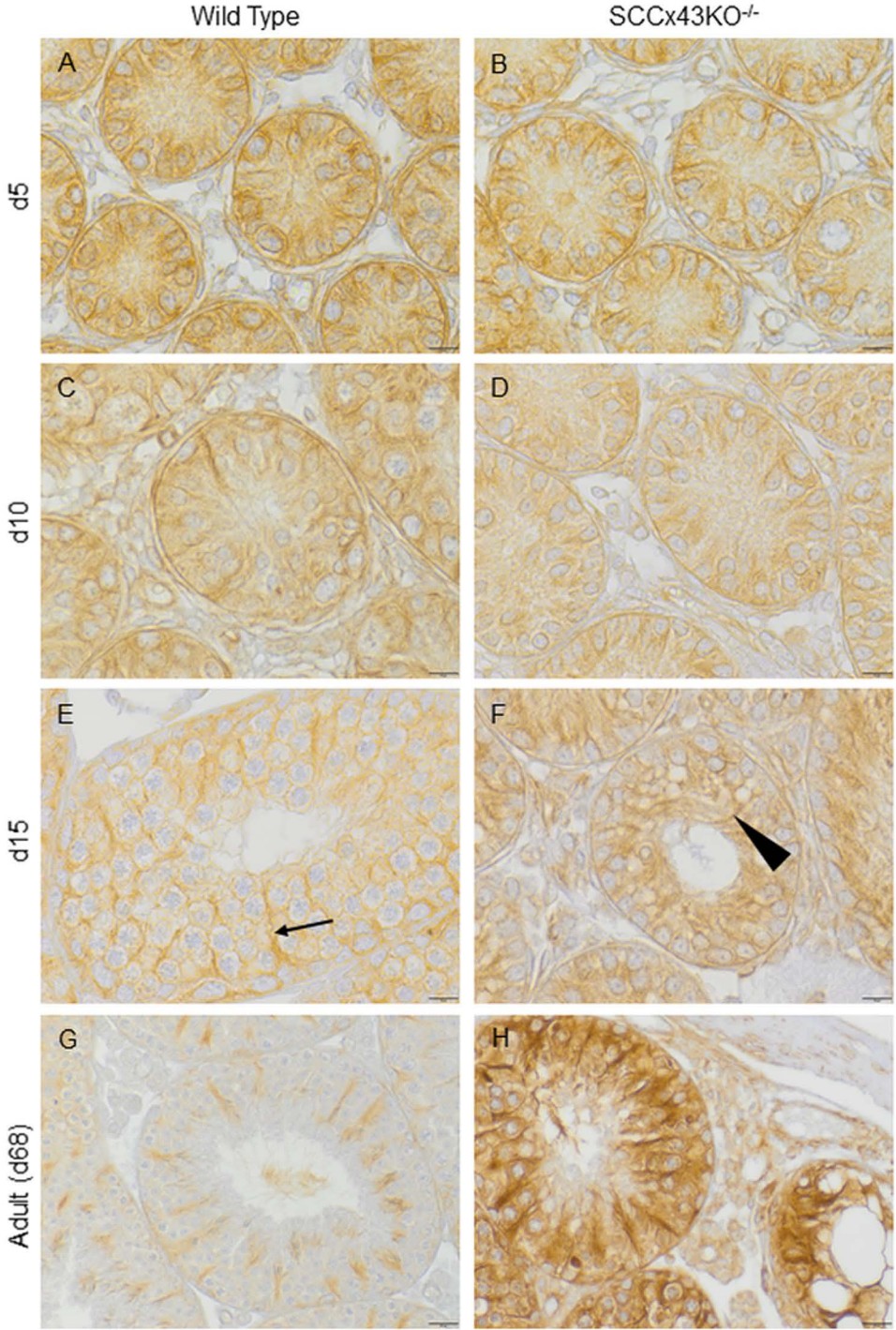

**Fig 3. Immunostaining for α-tubulin of wild type (A, C, E, G) and SCCx43KO⁻/⁻-mice (B, D, F, H) of different ages.** Exemplarily shown are samples of 5-, 10-, 15-, and 68-day old (adult) mice. In young mice (2-8 days postnatal), the Sertoli cell cytoplasm is diffusely immunopositive. No differences between both genotypes can be seen (A and B). Around day 10 postnatal, first faint denser structures appear in wild type seminiferous tubules (C) and become more evident in 12- and 15-day old mice (E). In wild type mice, those band-like structures are in line with the lumen (E, arrow), while in SCCx43KO⁻/⁻-mice denser "areas" appear later (after day 10) and are smaller, multitudinous and deviate to one side or another (F, arrowhead). With increasing age, wild type mice develop the characteristic spoke-like staining pattern in seminiferous tubules (E, G), while in knockout mice the denser

structures resemble the frayed ends of ropes (F). Furthermore, in knockout mice, spermatogenic arrest becomes even more evident (F) as well as the appearance of vacuoles within the tubule (H, right corner). However, some tubules of adult SCCx43KO⁻/⁻-mice show normal ("residual") spermatogenesis. Those tubules show a similar radial staining pattern as wild type tubules (H, left tubule); scale bars = 10 µm (A-F), 25 µm (G and H).

SC (Fig 3A and 3B). However, no differences in α- tubulin distribution had yet occured between both genotypes. At the age of 10 days, first faint cable-like structures became distinctive in WT mice (Fig 3C). In 12- and 15-day-old SCCx43KO⁻/⁻-mice, seminiferous tubules began to show band-like structures, but those were laterally deviated and not in line with the lumen. Their coeval WT littermates featured a radial staining pattern with straight "spokes" (Fig 3E and 3F). From day 18 onwards, SCCx43KO⁻/⁻-mice exhibited more and larger vacuoles within the germinal epithelium (Fig 3H, lower right corner). This was accompanied byan arrest of spermatogenesis at the level of spermatogonia and a lateral deviation of α-tubulin positive structures. The spoke-like staining pattern, which is characteristic for α-tubulin in WT mice, could not be found in KO mice (Fig 3F) (a complete presentation of all age groups is found in S3 Fig).

Astonishingly, adult mutant mice sometimes contained tubules with normal "residual" spermatogenesis. Those tubules immunostained in the same way as tubules of WT mice (Fig 3H, left upper corner). Due to the fact that microtubules are only 25 nm in diameter [45], it is fairly impossible to investigate the alignment and orientation of individual tubules in detail under a light microscopy. Therefore, testes samples of adult SCCx43KO⁻/⁻-mice and WT littermates were also investigated via TEM (Fig 4). With 21,560-fold magnification, microtubules became visible within the SC cytoplasm (Fig 4E and 4F). In both genotypes, microtubules were approximately parallel to each other. However, microtubules of Cx43 deficient SC were not always in line with the lumen but rather "followed" the deviating cell shape or the finger-like cell extensions (Fig 4D and 4F). The α-tubulin antibody was also used for immunogold labeling. The gold particles were evenly distributed throughout the cytoplasm of SC in both genotypes but did not allow the detailed tracking of individual microtubules (Fig 5A and 5B).

### 3.3. Actin filaments

Testis sections of adult WT mice are immunopositive for β-actin within the seminiferous epithelium near the basement membrane just above the spermatogonia (Fig 6A and 6C, arrow). Labelling was also found around the heads of spermatids (Fig 6A and 6C, arrowhead). Depending upon their developmental stage, those spermatids were located directly at the luminal edge or drawn deep into infoldings of the SC membrane. A similar staining pattern was described by Cavicchia et al. [46]. However, adult SCCx43KO⁻/⁻-mice showed a diffuse cytoplasmic staining pattern (Fig 6Band 6D), which also applied to intratubular cell clusters. In cases of residual spermatogenesis (the staining pattern of tubules with residual spermatogenesis for α-tubulin and β-actin is found in S4 Fig), labeling was found near the basement membrane and around spermatids in these tubules (Fig 6B, arrow and arrowhead).

Like microtubules, filamentous actin is formed by polymerization of monomeric "building blocks", in this case called G-actins [21]. For detection of filamentous actin, phalloidin was used. Phalloidin is a bicyclic heptapeptide derived from amanita phalloides, a poisonous mushroom [47] that binds to F-actin more tightly than to G-actin [48]. In adult WT mice, phalloidin IF (Fig 7) caused a similar staining pattern as the IHC with an antibody directed against β-actin: labeling was detected near the basement membrane and at sites of attachment between SC and spermatids (Fig 7A). The basally located labeling appeared at the same location as claudin-11 (Fig 7C and 7G), a well-known BTB component in mice and men [49–51]. In Cx43 deficient SC, phalloidin and claudin-11 caused a diffuse cytoplasmic staining pattern (Fig 7B, 7D and 7H). For claudin.11, this diffuse staining is known from IHC [50].

TEM allowed an even more defined investigation of actin filaments (Fig 8). Being the smallest component of the cytoskeleton with a diameter of only 7 nm [52], actin filaments are known parts of the basal and apical ES. There, these filaments are sandwiched between the endoplasmic reticulum and the plasma membrane of the SC [26]. Morphologically

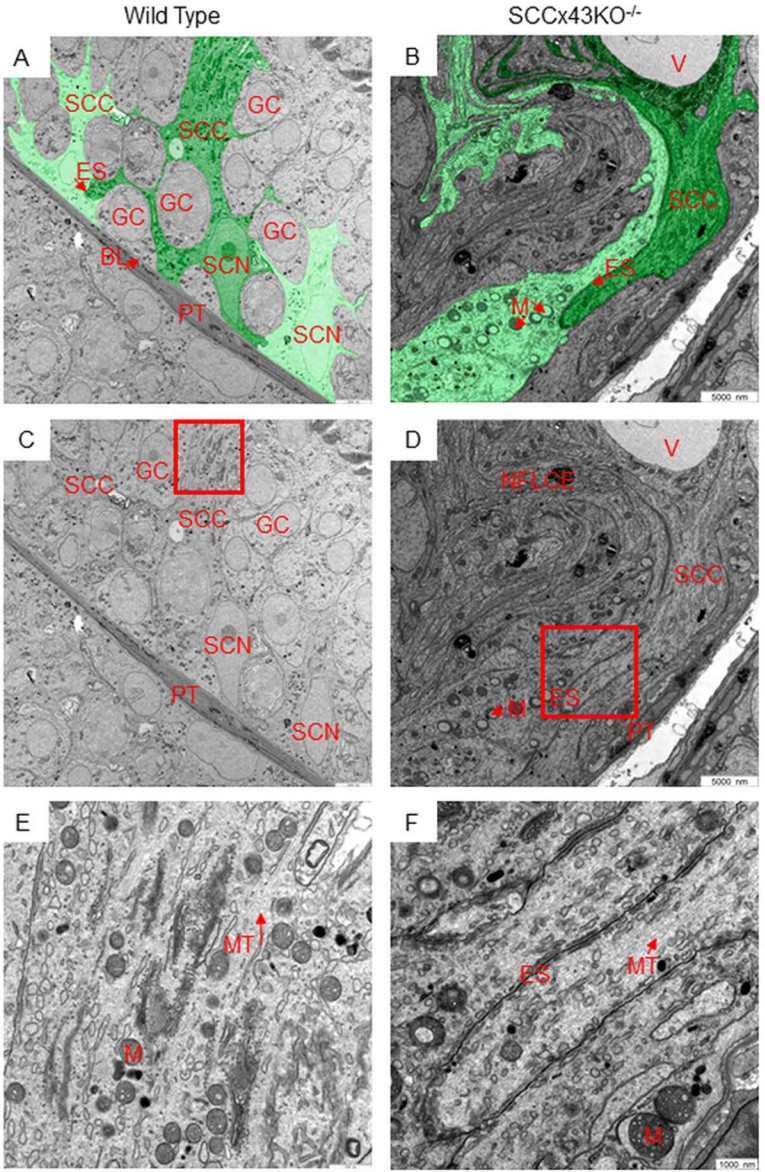

**Fig 4. Transmission electron microscopy of testes samples of adult wild type mice (A, C, E) and their SCCx43KO[-/-]-littermates (B, D, F).** The cytoplasm of individual Sertoli cells (A and B) is retroactively dyed green (different shades of green for different Sertoli cells) via image editing software (Gimp, GNU Image Manipulation Program). While wild type mice feature Sertoli cells with the characteristic stem like cell shape and extensions around germ cells of different developmental populations (A), Connexin 43 deficient Sertoli cells look like deflated balloons. They show numerous finger-like extensions, which intertwine with the extensions of adjacent Sertoli cells (B). The red boxes in C and D are shown in E and F in higher magnification (21,560-fold). In both genotypes, microtubules (E, F red arrows) are linearly positioned. In SCCx43KO[-/-]-mice, microtubules are more closely spaced. Please note that the microtubules in F are not in line with the apico-basal axis of the cell but rather "follow" its' aberrant cell shape. BL= basal lamina, ES= ectoplasmic specialisations, GC= germ cell, NFLCE= numerous finger-like cell extensions, M= mitochondrion, MT= microtubules, PT= peritubular cell, SCC= Sertoli cell cytoplasm, V= vacuole; scale bars = 5000 nm (A-D), 1000 nm (E and F).

intact basal ES could be found in mice of both genotypes (Fig 8C and 8D). In addition, these structures were positively immunolabeled using the aforementioned antibody directed against β-actin and a secondary, gold-labeled antibody (immunogold labeling, Fig 5E–5H). However, in SCCx43KO[-/-]-mice, basal ES seemed to be increased in number and in

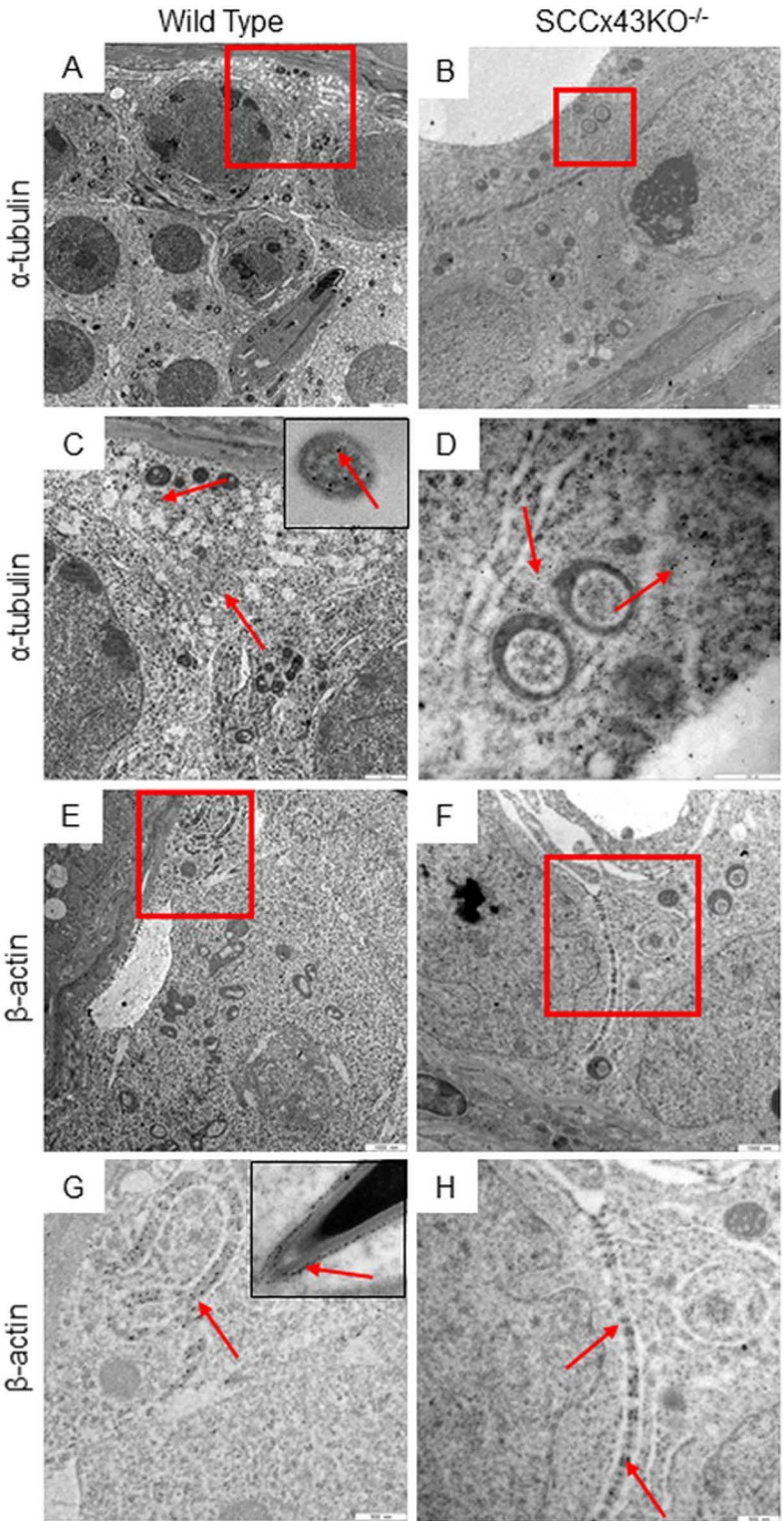

**Fig 5. Immunogold labeling for α-tubulin (A-D) and β-actin (E-H).** The red boxes in A, B, E and F are shown in a higher magnification in C, D, G, and H. In adult wild type (A and C) and SCCx43KO⁻/⁻-mice (B and D), immunolabelling indicates the existence of microtubules in the Sertoli cell cytoplasm. Gold particles can also be found at the central microtubule core of sperm flagella (C, insert), which were used as positive controls. However, the method

at hand did not allow the longitudinal tracking of individual microtubules in the Sertoli cell cytoplasm. Nonetheless, the antibody directed against -actin caused a very accurate labeling of what can be considered as apical (G, insert) and basal ectoplasmic specializations (E-H). Again, the significantly enlarged spatial appearance of basal ectoplasmic specializations in mutant mice becomes evident; scale bars = 2500 nm (A), 1000 nm (B, E, and F), 500 nm (C, D, G, and H).

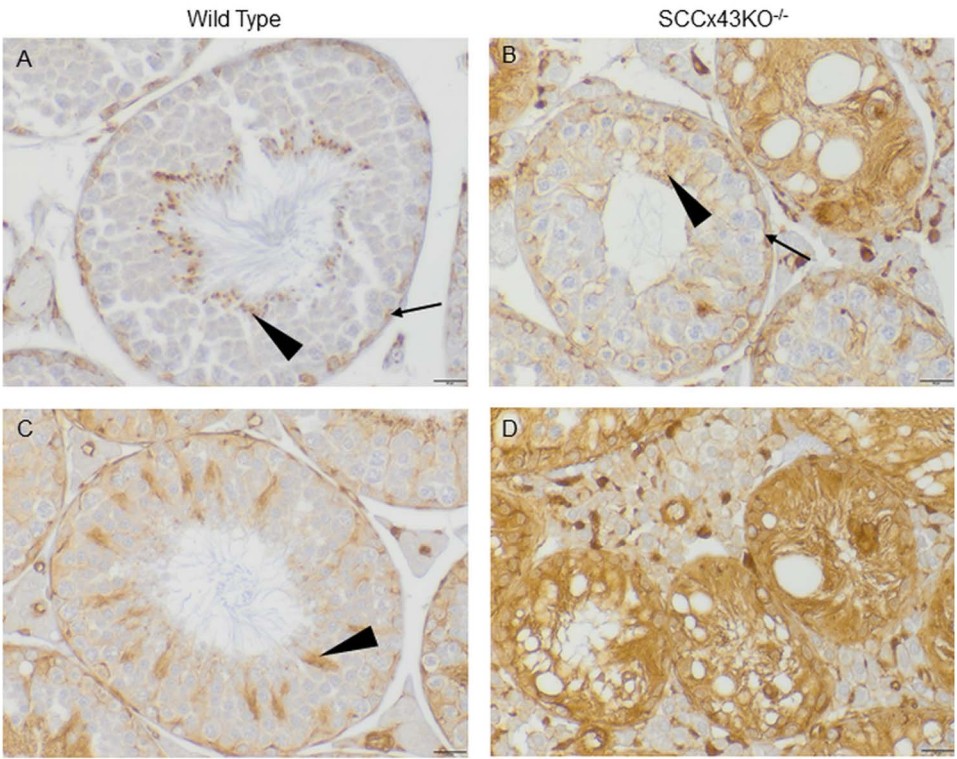

**Fig 6. Immunostaining for β-actin.** In adult wild type mice (A and C), β-actin is located near the basement membrane (A, arrow). Depending upon the epithelial stage, it can also be observed around the heads of spermatids at the luminal edge or in the deep crypts of the Sertoli cell (A and C, arrowhead). In adult mutants (B and D), β-actin seems to be rather diffusely distributed throughout the Sertoli cell cytoplasm (D). Occasionally, even SCCx43KO$^{-/-}$-mice feature some tubules with rather normal spermatogenesis. Those tubules show a similar, but finer, staining pattern than the wild type littermates with β-actin around spermatids (B, arrowhead) and are basally located (B, arrow); scale bars = 20 µm.

their spatial extension (Fig 5D and 5F). Basal ES stretched out relatively wide towards apical cell regions, while the basal ES of WT mice were restricted to a small area near the basement membrane. Since Cx43 deficient seminiferous tubules are (in most cases) unable to support spermatogenesis and feature an arrest of the latter at the level of spermatogonia, mutant SC also did not exhibit apical ES.

### 3.4. Intermediate filaments

Consistent with previous studies [14], vimentin was detectable in SCCx43KO$^{-/-}$-mice and their WT littermates from day two postnatal up to adulthood. In immature SC, vimentin was accumulated basally of the SC nucleus (Fig 9A and 9B), whereas in adult SC, it was localized around the SC nucleus, "sending out" apical extensions [20]. In most cases, the apical vimentin extensions were shorter in SCCx43KO$^{-/-}$-SC (Fig 9D) compared to their WT littermates (Fig 9C). SC that formed intratubular cell clusters in Cx43 deficient seminiferous tubules also stained immunopositive for vimentin (Fig 9D).

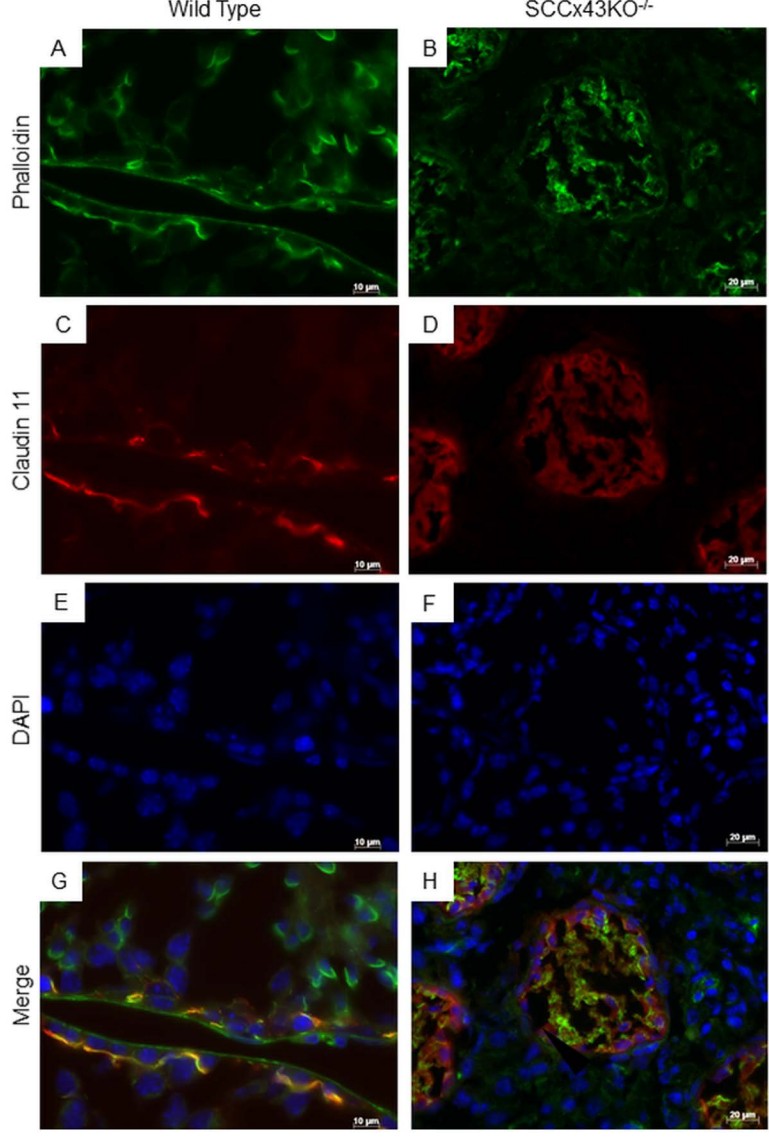

**Fig 7. Immunofluorescence of adult SCCx43KO⁻/⁻-mice (B, D, F, and H) and coeval wild type littermates (A, C, E, and G) labeled for actin (phal-loidin/ green), the blood-testis-barrier protein claudin-11 (red) and DNA (DAPI/ blue).** In wild type mice, phalloidin stains actin filaments near the base of the seminiferous tubule at the same location where, according to the claudin-11 positive signal, the blood-testis-barrier can be found (A, C, and G). Additionally, actin filaments are arranged at sites of attachment between Sertoli cells and spermatids (A). In SCCx43KO⁻/⁻-mice, however, phalloidin and claudin-11 are diffusely distributed within the cytoplasm of the Sertoli cell (B, D and H); scale bars = 10 μm (A, C, E, G), 20 μm (B, D, F, H).

TEM permitted the inspection of those filaments in more detail (S5 Fig). Intermediate filaments could be detected around SC nuclei in both genotypes.

None of the seminiferous tubules of any age and phenotype stained immunopositive for keratins neither within the seminiferous epithelium nor in the intratubular clusters (Fig 10A, 10B, 10D and 10E). However, both antibodies used for the detection of keratins caused an immunopositive reaction in the simple columnar epithelium and glandular cells of the uterus [53] (Fig 10C), in the epidermis, sebaceous and sweat glands, and the epithelial hair root sheath, epithelia of bile ducts and renal tubules (which were used as positive controls, but are not shown) as described by several authors

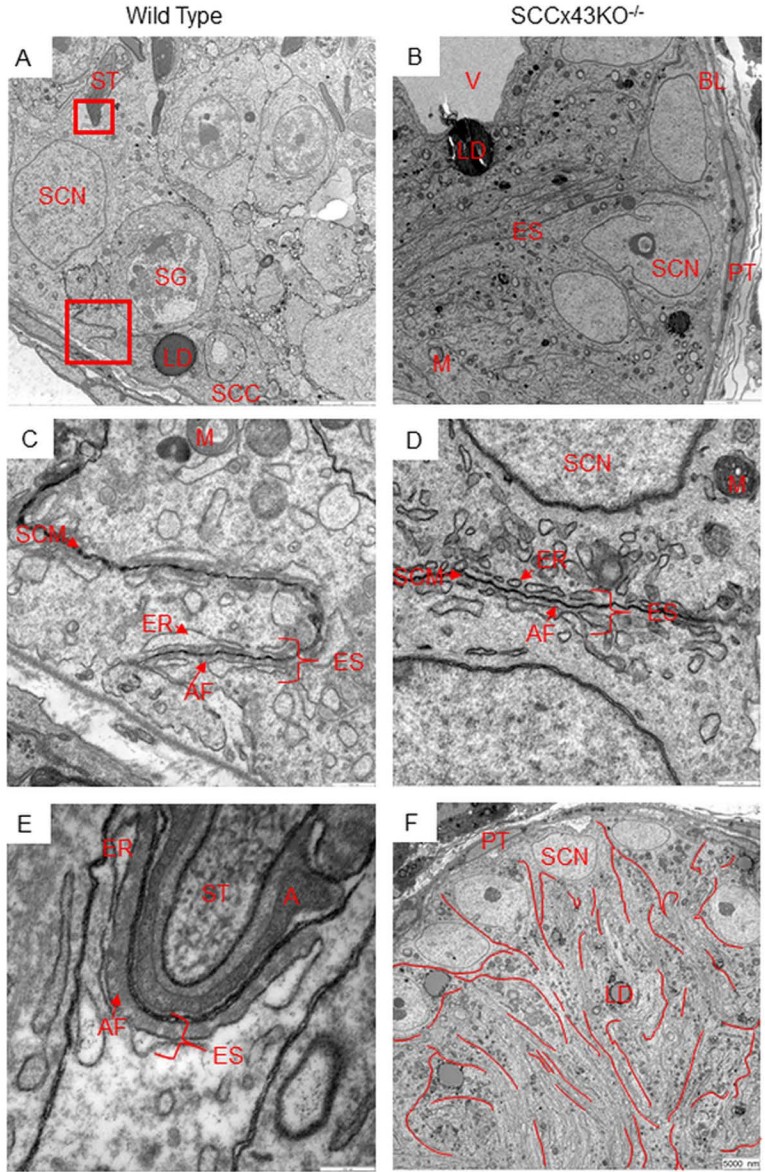

**Fig 8. Transmission electron microscopy of testes samples of adult SCCx43KO<sup>-/-</sup>-mice (B, D, F) and their wild type littermates (A, C, E).** The red boxes in A and B are shown in higher magnification in C and E or in D. In both genotypes, basal ectoplasmic specializations can be found near the base of the epithelium (red boxes in A and B) and feature the same structure (C and D). In mutants, there seem to be more basal ectoplasmic specializations (F). They even stretched out towards relatively apical cell regions (B, the ectoplasmic specializations extend nearly across the entire width of the image). In F, the basal ectoplasmic specializations are retraced in red via image editing software (Gimp, GNU Image Manipulation Program). In wild type mice, apical ectoplasmic specializations are situated at sites of attachment between Sertoli cells and spermatids (E). Since the SCCx43KO<sup>-/-</sup>-mice feature a spermatogenic arrest at the level of spermatogonia, they exhibit neither spermatids nor apical ectoplasmic specializations. AF= actin filaments, BL= basal lamina, ER= endoplasmic reticulum, ES= ectoplasmic specializations, GC= germ cell, LD= lipid droplet, NFLCE= numerous finger-like cell extensions, M= mitochondrion, MT= microtubules, PT= peritubular cell, SCC= Sertoli cell cytoplasm, SCN= Sertoli cell nucleus, SCM= Sertoli cell membrane, SG= spermatogonia, ST= spermatid, V= vacuole; scale bars = 5000 nm (A, B, and F), 500 nm (C and D), 250 nm (E).

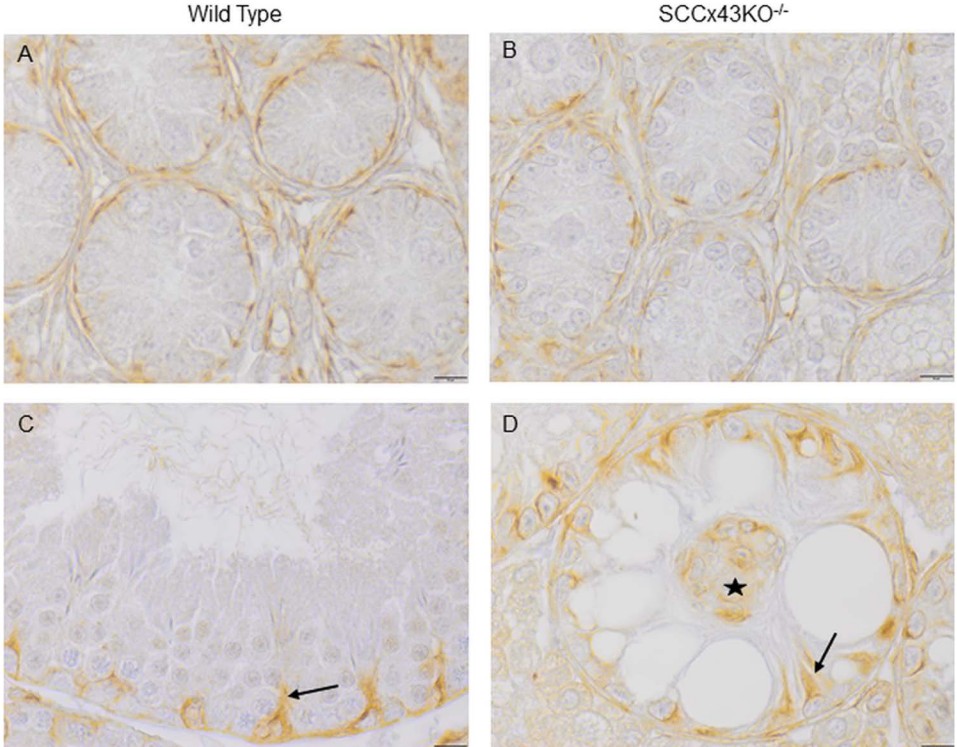

**Fig 9. Immunolabelling of vimentin at different ages.** Exemplarily shown are sections of 2-day (A and B) and 68-day-old (C and D) wild type (A and C) and SCCx43KO$^{-/-}$-mice (B and D). Sertoli cells of all investigated ages and both genotypes stained immunopositive for vimentin. Confirming previous results, its distribution changed from an accumulation basal of the Sertoli cell nuclei in immature Sertoli cells (A and B) to a perinuclear localization with apical extensions (arrows in C and D). These extensions were found to be mostly shorter in SCCx43KO$^{-/-}$- mice. Intratubular cell clusters formed by Sertoli cells (marked by the star in D) are one of the characteristic morphological features of mature SCCx43KO$^{-/-}$ seminiferous tubules. These clusters of attached cells stained immunopositive for vimentin and showed smaller nuclei with heterochromatic patches; scale bars = 10 μm.

[54–56]. Furthermore, the second antibody caused a staining pattern in the cell periphery of hepatocytes typical for the keratins 18 and 8 [56–58] (Fig 10F).

## 4. Discussion

The SC specific KO of Cx43 mainly leads to an arrest of spermatogenesis at the level of spermatogonia [7,8]. The present study shows that this loss of Cx43 comes along with dramatic changes of the cell shape and the cytoskeletal structure of the somatic SC.

### 4.1. Microtubules

Adult Cx43 deficient SC seem to collapse like deflated balloons and feature vacuoles between SC in many cases or even within the SC cytoplasm. These cell shape alterations indicate disturbances of the structures responsible for keeping the former shape. Microtubules are known to be one of these responsible structures [20,59], if not the most important ones. When microtubule polymerization is blocked via colchicine, GC, and parts of the SC cytoplasm slough [19,59,60], indicating that microtubules are important for the integrity of the seminiferous epithelium. It has been presumed that those treated cells lose cytoskeletal support, "round up" and that apical cell extensions would break from the cell body [59]. Since Cx43 can directly interact with α- and β-tubulin [61–63], and next-generation sequencing of 8-, 10- and 12-day old

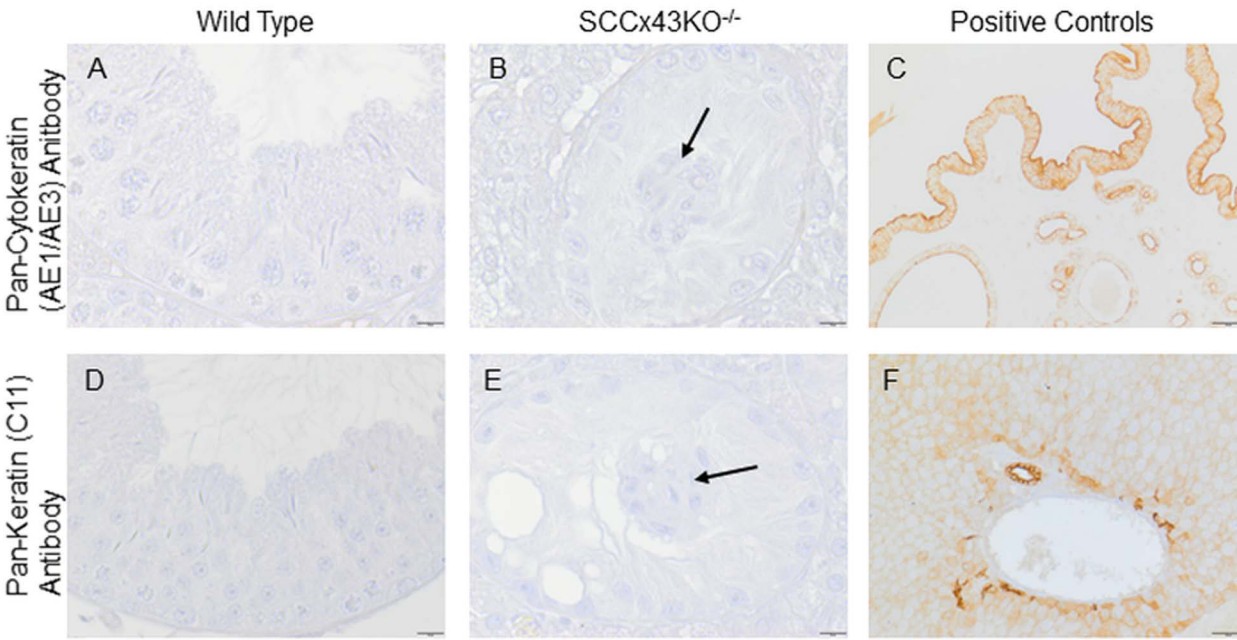

**Fig 10. Immunolabeling with the pan-cytokeratin antibody (AE1/AE3) that binds to keratin 1-8, 14-16, and 19 (A, B, C), and the pan-keratin (C11) monoclonal antibody that detects keratin 4-6, 8, 10, 13, and 18 (D, E, F).** The sections are gently counterstained with hematoxylin. None of the sections of any age and phenotype stained immunopositive for keratins including the cluster forming Sertoli cells (arrows in B and E). Exemplarily shown are sections of adult wild type (A and D) and SCCx43KO$^{-/-}$ mice (B and E). As positive controls (among others), sections of murine uterus (C) and murine liver (F) were used; scale bars = 10 μm (A, B, D, E), 50 μm (C and F).

SCCx43KO$^{-/-}$-mice revealed a significant down-regulation of several genes (*Tuba3a, Tuba3b, Tuba4a,* and *Tubb4b*) that encode for isoforms of α- and β-tubulin [39], an altered microtubule distribution in Cx43 deficient SC was conceivable.

IHC, TEM, and immunogold labeling indicated that Cx43 deficient SC feature microtubules. Using IHC, the entire cytoplasm of SC seemed to be immunopositive for α- and β-tubulin, or labeling appeared as many dense structures that are reminiscent of the frayed ends of ropes. However, it must not be overlooked that in Cx43 deficient seminiferous tubules, GC are predominantly absent. In contrast, each WT SC sustains and nurtures a certain number of GC [64] that account to a large degree for the seminiferous epithelial volume [65]. Hence, the WT SC cytoplasm appears trunk-like and immunolabeling for α-tubulin within its cytoplasm resembles spokes. During different stages of spermatogenesis, microtubule organization (and the SC shape) changes concomitantly with the altered position of spermatids. Hence, immunolabeled tubulin appears like a thick band when elongated spermatids are embedded in the SC deep crypts, and more like thin cables that apically branch out when late spermatids are moved to the luminal edge of the epithelium [44]. In SCCx43KO$^{-/-}$-mice, only tightly packed SC (and single spermatogonia) make up the seminiferous epithelium. Hence, SC are tightly packed together and the seminiferous tubules appear to be reduced in diameter. This could be one explanation for the immunostaining of virtually the entire seminiferous tubule. The resemblance to denser structures reminiscent of frayed ends of ropes may arise from the "finger-like cell extensions" (the peripheral cuts of the extraordinary SC shape visible in 2D imaging) of adjacent SC that intertwine. Considering that some SC resemble a wave, these altered cell shapes may explain the impression of frayed ends of ropes that are not in line with the lumen but deviate to one or the other side.

TEM confirmed that Cx43 deficient SC can form microtubules instead of enclosing only separate and unpolymerized subunits. Microtubules were straight, linearly positioned, and parallel to each other. Astonishingly, the microtubules were parallel to each other but not necessarily oriented along the apico-basal axis of the cell. Rather, microtubules "followed"

the aberrant SC shape or finger-like cell extensions. A misorientation of microtubules being causative for the altered cell shape could be presumed. Microtubules are linked to the cisternae of the apical ES [66]. Consequently, a lack of apical ES may cause the observed lack of microtubular orientation. Considering that SCCx43KO$^{-/-}$-mice show characteristic radial α- and β-tubulin staining pattern in tubules with residual spermatogenesis, this theory becomes more plausible. A lack of Cx43 in those tubules has been verified via β-galactosidase IHC in the past [50].

Additionally, the immunostaining pattern of α-tubulin in SCCx43KO$^{-/-}$-mice starts to differ from those of coeval WT littermates concomitant with "non-appearance" or lack of differentiating GC during puberty. In prepubertal mice, immuno-labelling for α- and β-tubulin is similar. At the age of 2 days postnatal, murine seminiferous epithelium consists of SC and gonocytes. In 5-day-old mice, the gonocytes differentiate into type A spermatogonia that are located on the edge of the basement membrane [67]. Starting around day 10, not only first spermatocytes could be found [67], but also immunola-beling for α-tubulin in SCCx43KO$^{-/-}$-mice started to differ. The cable-like "structures" were fainter and started to form later in mutant tubules. With increasing age, it became more and more obvious that the denser structures in KO mice were not in line with the lumen. Simultaneously, the substantial absence of GC in mutant mice became evident [68]. This correla-tion of progressive GC deficiency and altered microtubule distribution may also hint at the lack of GC as causative for the observed alterations of the staining pattern rather than direct influence of lacking Cx43.

In some studies of other KO mouse models that genetically induced impaired spermatogenesis, disturbance of micro-tubule distribution has been noticed as well [69–73]. However, in those cases, spermatogenesis was not disturbed at the level of spermatogonia but later in GC development or the attachment of GC was faulty, making the drawing of explana-tory parallels impossible.

To date, the question whether Cx43 influences microtubule organization directly via interaction/binding, or indirectly via lack of GC that leads to disorganization, cannot be answered conclusively. However, the "normal" staining pattern of mutant but not GC devoid tubules and the correlation of the beginning GC deficiency with an altered staining pattern in growing mice provide a strong indication of the latter.

## 4.2. Actin filaments

In adult WT mice, the antibody directed against β-actin (and phalloidin as well) caused immunopositive reactions at the level of the BTB (at the same localisation as BTB component claudin11) and around the heads of spermatids. Hence, actin filaments occured, where basal and apical ES are located. Adult SCCx43KO$^{-/-}$-mice showed a rather diffuse staining pattern. Here too, a lack of GC may offer one possible explanation for this altered, extensive immunopositive signal. TEM revealed that Cx43 deficient SC feature intact basal ES. Actually, ES extended to relatively apical cell membrane regions and thereby appeared to have a far larger surface area than in WT mice. Previous ultrastructural analysis of Cx43 deficient SC revealed an increased number of tight and adherens junctions per unit length of the BTB as well. IHC and Western blot analysis confirmed an increased amount of numerous junction proteins (e.g., N-cadherin, β-catenin and occludin) in SCCx43KO$^{-/-}$-mice. Furthermore, the formation of a BTB was confirmed via electron opaque tracer (lanthanum nitrate) and hypertonic fixation solution (glucose) [13]. Interestingly, in those functional studies, the BTB in mutant mice seemed unaltered or even slightly tighter than in WT lit-termates, leading to the assumption that Cx43 may not be crucial for establishing a functional BTB. Rather, it seems to interfere with BTB dynamics (assembly and disassembly) [12]. Additionally, Cx43 also seems to have an impact on basal ES formation and localization as well, resulting in a diffuse double immunostaining pattern for phalloidin and claudin-11 in Cx43 deficient SC. Considering that basal ES are intermingled with tight junctions [74,75], this observation comes as no surprise.

Since tubules of SCCx43KO$^{-/-}$-mice with residual spermatogenesis showed positive immunolabeling at sites of attach-ment to spermatids and at sites of BTB, the influence of the formation of apical and basal ES by Cx43 is in all likelihood an indirect one. While ultimate and clarifying proof has still not been given, comparison to a mouse model with W locus (white spotting gene) mutation supports this theory. W/W$^v$ mice feature GC deficiency due to a failure of GC progeni-tors to migrate to the gonadal ridge during embryological development [76,77]. Alongside smaller and highly vacuolated

seminiferous tubules, those mutant mice possess more basal ES in comparison to WT mice while apical ES were lacking [78]. Extensive basal ES are also known from men with SCOS [79], germinal aplasia, or severe germ cell depletion [80], hinting at a correlation of missing GC and ES formation.

### 4.3. Intermediate filaments

The role of intermediate filaments is still unclear. Its generalized loss does not have an obvious impact on spermatogenesis or maturity [81]. However, different cell types feature intermediate filaments of different proteins. The large family of intermediate filaments can be subdivided into six groups (type I-VI) [54]. Hence, those proteins can be used as marker for identifying the origin, differentiation status and type of cells. Vimentin, a member of group III, is found in cells of mesenchymal origin, while keratins can be found in epithelial cells [82]. Anyhow, cells may express more than one intermediate filament protein and the expression of intermediate filament proteins can be developmentally regulated [83]. While in fetal and early new-born testes of rats vimentin and keratins are coexpressed [31], SC of adult mice express only intermediate filaments of the vimentin type [34]. Reexpression of keratins (chiefly keratin 8 and 18) occurs in seminiferous tubules of men with spermatogenic impairment, such as spermatogenic arrest or SCOS as well as in SC of GC neoplasia in situ infiltrated tubules plus in senile SC [4,37]. Since there is a correlation of Cx43 and disturbed spermatogenesis, seminoma and the GC neoplasia in situ [2–5], and next-generation sequencing of murine SCCx43KO$^{-/-}$-SC has detected an upregulation of the *Krt 18* gene in prepubertal KO mice [39], a re-expression of keratins in our mutant mice was conceivable. This altered intermediate filament synthesis/expression would have hinted at immaturity or dedifferentiation of Cx43 deficient SC. Vimentin filaments were centered around the SC nucleus and could be found in both genotypes via TEM and IHC. Confirming previous results [14], vimentin localization changed from being basal of the SC nucleus in immature SC to a perinuclear congregation with apical extensions. Those extensions were mostly shorter in KO mice than in their WT littermates. Apart from that, no differences were apparent between the two genotypes. Keratins were not detectable in SC in any genotype of any observed age. This leads to the conclusion that loss of Cx43 seems to have an impact on vimentin distribution, but leads neither to an altered keratin synthesis nor to a reexpression/resynthesis of keratins in SC.

### 5. Concluding remarks

Loss of Cx43 in murine SC is accompanied by dramatic changes of its cell shape, microtubule organization, the spreading of basal ES along the SC membrane, and the absence of apical ES in mature SC. Whether those changes are caused directly by lacking interaction with Cx43 or indirectly by the lack of GC (caused by Cx43 deficiency) remains unclear (with tendencies towards the latter). Filament polymerization and ES assembly as well as intermediate filament synthesis seem not to be influenced by Cx43 or its loss. Furthermore, the missing GC in SCCx43KO$^{-/-}$-mice might not only be explained by alteration of the cytoskeleton, albeit reciprocity and some additional hampering influence cannot be entirely ruled out.

### Supporting information

**S1 Fig. Hematoxylin and Eosin (H&E) stained testis sections (fixed with Bouin's solution and embedded in paraffin) of an adult wild type mouse (A) and an adult SCCx43KO$^{-/-}$-mouse (B).** Wild type mice feature seminiferous tubules with normal spermatogenesis. However, Connexin 43 deficient tubules exhibit an arrest of spermatogenesis at level of spermatogonia or Sertoli cell-only syndrome (SCOS). Furthermore, in SCCx43KO$^{-/-}$-mice, the seminiferous tubules are highly vacuolated (star) and feature intratubular cell cluster (arrow); scale bars = 20 μm.
(TIF)

**S2 Fig. Negative controls for α-tubulin, β-actin, and vimentin immunohistochemistry (A, B, E, F, I, and J) and immunogold labeling (C, D, G, H, K, and L).** Scale bars = 20 μm (A, B, E, F, I, and J), 500 nm (C, D, G, H, K, and L).
(TIF)

**S3 Fig. Immunostaining for α-tubulin of wild type and SCCx43$^{-/-}$-mice of different age groups.**
(TIF)

**S4 Fig. Immunohistochemistry (not counterstained) for α-tubulin (A and B) and β-actin (C and D) of adult wild type mice (A and C) and adult SCCx43KO$^{-/-}$-mice with residual spermatogenesis (B and D).** The antibody against α-tubulin causes a radial staining pattern in wildtype mice and in those tubules of SCCx43KO$^{-/-}$-mice which feature residual spermatogenesis. Immunolabeling for β-tubulin in wild type mice can be found near the basement membrane just above the spermatogonia as well as around elongating spermatids. While most Connexin 43 deficient tubules immunostained diffusely positive for β-actin, the tubules with residual spermatogenesis show an analogical "normal" staining pattern; scale bars = 20 μm.
(TIF)

**S5 Fig. Transmission electron microscopy of testes samples of adult SCCx43KO$^{-/-}$-mice.** The red boxes are shown in higher magnification in B and C. Intermediate filaments (IF) surround the Sertoli cell nucleus (SCN); scale bars = 5000 nm (A), 1000 nm (B), 250 nm (C).
(TIF)

**S1 File. Video sequence of the seminiferous epithelium of an adult wild type mouse observed via SBF-SEM.** The cytoplasm of a Sertoli cell is colored pink, its nucleus brown, and the adjacent germ cells green and light blue for better visualization. The cytoplasm of a second Sertoli cell is dyed dark blue and its nucleus brown.
(ZIP)

**S2 File. Video sequence of the seminiferous epithelium of an adult SCCx43KO$^{-/-}$-mouse observed via SBF-SEM.** The cytoplasm of a Sertoli cell is colored pink, its nucleus brown. Note that 2D imaging gives the impression of several finger-like cell extensions that in 3D reconstruction turned out to be peripheral cuts of the extraordinary cell shape of the mutant Sertoli cell.
(ZIP)

**S3 File. Video sequence of a 3D reconstructed wild type Sertoli cell.**
(ZIP)

**S4 File. Video sequence of a 3D reconstructed SCCx43KO$^{-/-}$- Sertoli cell (1).**
(ZIP)

**S5 File. Video sequence of a 3D reconstructed SCCx43KO$^{-/-}$- Sertoli cell (2).**
(ZIP)

**S6 File. Video sequence of a 3D reconstructed SCCx43KO$^{-/-}$- Sertoli cell (3).**
(ZIP)

## Acknowledgments

The authors are also grateful for the support and advice of Rex A. Hess.

## Author contributions

**Conceptualization:** Kristina Rode, Ralph Brehm.

**Data curation:** Mareike Ueffing.

**Methodology:** Marion Langeheine.

**Project administration:** Ralph Brehm.

**Resources:** Marion Langeheine, Sarah Staggenborg, Gudrun Wirth, Kerstin Rohn, Rüdiger Koch, Ines Blume, Christiane Pfarrer, Christoph Wrede.

**Supervision:** Ralph Brehm.

**Writing – original draft:** Mareike Ueffing.

**Writing – review & editing:** Sarah Gniesmer.

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
