## [Decision Letter · Decision Letter 0]

7 Oct 2024

PONE-D-24-29399The impact of Connexin 43 deficiency on the cell shape and cytoskeleton of murine Sertoli cells: A house with ramshackle walls?PLOS ONE

Dear Dr. Gniesmer,

Thank you for submitting your manuscript to PLOS ONE. After careful consideration, we feel that it has merit but does not fully meet PLOS ONE’s publication criteria as it currently stands. Therefore, we invite you to submit a revised version of the manuscript that addresses the points raised during the review process. The manuscript has to be thoroughly revised to avoid grammatical errors. On the experimental side, Provide further evidence on the absence of the radial spike like organization in the knock out mice. Co-staining of the Sertoli cells with a specific marker is anticipated.

We look forward to receiving your revised manuscript.

Kind regards,

Suresh Yenugu

Academic Editor

PLOS ONE

Journal Requirements:

2. We noted in your submission details that a portion of your manuscript may have been presented or published elsewhere. Our article contains the results of primary scientific research, which was also published as a dissertation in June 2023 (Ueffing, 2023, Auswirkungen des Sertoli-Zell-spezifischen Verlusts des Gap-Junction Proteins Connexin 43 auf die Zellgestalt und das Zytoskelett von Sertoli, Monographie, Tiermedizin, Tierärztliche Hochschule Hannover), Please clarify whether this publication was peer-reviewed and formally published. If this work was previously peer-reviewed and published, in the cover letter please provide the reason that this work does not constitute dual publication and should be included in the current manuscript.

Reviewers' comments:

Reviewer's Responses to Questions

**Comments to the Author**

1. Is the manuscript technically sound, and do the data support the conclusions?

Reviewer #1: Yes

Reviewer #2: Yes

2. Has the statistical analysis been performed appropriately and rigorously? 

Reviewer #1: I Don't Know

Reviewer #2: Yes

3. Have the authors made all data underlying the findings in their manuscript fully available?

Reviewer #1: Yes

Reviewer #2: Yes

4. Is the manuscript presented in an intelligible fashion and written in standard English?

Reviewer #1: Yes

Reviewer #2: Yes

5. Review Comments to the Author

Reviewer #1: The manuscript submitted by Mareike et al., present a good piece of work. Written in simple english without difficult scientific jargon.

1. The figure 3 shows the age specific progression of alpha Tubulin organisation in wild type vs Sertoli cell specific CX3 knockout mice. It looks like the gap junction protein helps in acting as anchor for the radial spike like organisation of the tubulin protein or is it that the lack of germ cells in the knockout mice, creates a vacuole in the germinal epithelium. Due to which the organisation of radial spike like arrangement is absent? Please comment on this. Will co-staining of alpha tubulin and cx43 in wild type lead to better understanding?

2. In the figure 7, a co-staining of the Sertoli cell marker like sox9 along with phalloidin / claudin 11 will give a better picture of the claim in line 423, that “phalloidin and claudin 11 are diffusely distributed within the cytoplasm of the Sertoli cell”.

Reviewer #2: this manuscript is well-presented. i see some grammatical error.

this manuscript is well-presented. i see some grammatical error.

6. PLOS authors have the option to publish the peer review history of their article (what does this mean? ). If published, this will include your full peer review and any attached files.

**Do you want your identity to be public for this peer review?** For information about this choice, including consent withdrawal, please see our Privacy Policy .

Reviewer #1: No

Reviewer #2: **Yes: ** Hossein Azizi

---

## [Author Response · Author response to Decision Letter 1]

17 Feb 2025

The corresponding responses to the reviewers' comments can be found in detail and with images in the document “Response to Reviewers”.

---

## [Decision Letter · Decision Letter 1]

4 Mar 2025

The impact of Connexin 43 deficiency on the cell shape and cytoskeleton of murine Sertoli cells: A house with ramshackle walls?

PONE-D-24-29399R1

Dear Dr. Gniesmer,

We’re pleased to inform you that your manuscript has been judged scientifically suitable for publication and will be formally accepted for publication once it meets all outstanding technical requirements.

Kind regards,

Suresh Yenugu

Academic Editor

PLOS ONE

Additional Editor Comments (optional):

Reviewers' comments:

Reviewer's Responses to Questions

**Comments to the Author**

1. If the authors have adequately addressed your comments raised in a previous round of review and you feel that this manuscript is now acceptable for publication, you may indicate that here to bypass the “Comments to the Author” section, enter your conflict of interest statement in the “Confidential to Editor” section, and submit your "Accept" recommendation.

Reviewer #1: All comments have been addressed

Reviewer #2: (No Response)

2. Is the manuscript technically sound, and do the data support the conclusions?

Reviewer #1: Yes

Reviewer #2: (No Response)

3. Has the statistical analysis been performed appropriately and rigorously? 

Reviewer #1: Yes

Reviewer #2: (No Response)

4. Have the authors made all data underlying the findings in their manuscript fully available?

Reviewer #1: Yes

Reviewer #2: (No Response)

5. Is the manuscript presented in an intelligible fashion and written in standard English?

Reviewer #1: Yes

Reviewer #2: (No Response)

6. Review Comments to the Author

Reviewer #1: The author answered the queries raised with additional data. The co staining experiments increased the significance of the data substantially.

Reviewer #2: accepted

Please use the space provided to explain your answers to the questions above. You may also include additional comments for the author, including concerns about dual publication, research ethics, or publication ethics. (Please upload your review as an attachment if it exceeds 20,000 characters) (Limit 100 to 20000 Characters)

7. PLOS authors have the option to publish the peer review history of their article (what does this mean? ). If published, this will include your full peer review and any attached files.

**Do you want your identity to be public for this peer review?** For information about this choice, including consent withdrawal, please see our Privacy Policy .

Reviewer #1: No

Reviewer #2: No

---

## [Editor Report · Acceptance letter]

PONE-D-24-29399R1

PLOS ONE

Dear Dr. Gniesmer,

I'm pleased to inform you that your manuscript has been deemed suitable for publication in PLOS ONE. Congratulations! Your manuscript is now being handed over to our production team.

Kind regards,

on behalf of

Dr. Suresh Yenugu

Academic Editor

PLOS ONE